# First Report of the Prevalence at Baseline and after 1-Year Follow-Up of Treatable Traits in Interstitial Lung Diseases

**DOI:** 10.3390/biomedicines12051047

**Published:** 2024-05-09

**Authors:** Francesco Amati, Anna Stainer, Giacomo Maruca, Maria De Santis, Giuseppe Mangiameli, Chiara Torrisi, Paola Bossi, Veronica Polelli, Francesco Blasi, Carlo Selmi, Giuseppe Marulli, Luca Balzarini, Luigi Maria Terracciano, Roberto Gatti, Stefano Aliberti

**Affiliations:** 1Department of Biomedical Sciences, Humanitas University, Via Rita Levi Montalcini 4, Pieve Emanuele, 20072 Milan, Italy; anna.stainer@hunimed.eu (A.S.); maria.de_santis@hunimed.eu (M.D.S.); giuseppe.mangiameli@hunimed.eu (G.M.); paola.bossi@humanitas.it (P.B.); veronica.polelli@humanitas.it (V.P.); carlo.selmi@hunimed.eu (C.S.); giuseppe.marulli@hunimed.eu (G.M.); luca.balzarini@humanitas.it (L.B.); luigi.terracciano@hunimed.eu (L.M.T.); roberto.gatti@hunimed.eu (R.G.); stefano.aliberti@hunimed.eu (S.A.); 2IRCCS Humanitas Research Hospital, Respiratory Unit, Via Manzoni 56, Rozzano, 20089 Milan, Italy; 3Department of Pathophysiology and Transplantation, Università degli Studi di Milano, 20122 Milan, Italy; giacomo.maruca@unimi.it (G.M.); francesco.blasi@unimi.it (F.B.); 4Fondazione IRCCS Ca’ Granda Ospedale Maggiore Policlinico, Respiratory Unit and Cystic Fibrosis Adult Center, 20122 Milan, Italy; 5Rheumatology and Clinical Immunology, IRCCS Humanitas Research Hospital, Via Manzoni 56, Rozzano, 20089 Milan, Italy; 6Division of Thoracic Surgery, IRCCS Humanitas Research Hospital, Via Manzoni 56, Rozzano, 20089 Milan, Italy; 7Department of Diagnostic and Interventional Radiology, IRCCS Humanitas Research Hospital, Via Manzoni 56, Rozzano, 20089 Milan, Italy; chiara.torrisi@humanitas.it; 8Pathology Department, IRCCS Humanitas Research Hospital, Via Manzoni 56, Rozzano, 20089 Milan, Italy; 9Physiotherapy Unit, IRCCS Humanitas Research Hospital, Via Manzoni 56, Rozzano, 20089 Milan, Italy

**Keywords:** interstitial lung diseases, phenotypes, endotypes, treatable traits

## Abstract

Different factors, not limited to the lung, influence the progression of ILDs. A “treatable trait” strategy was recently proposed for ILD patients as a precision model of care to improve outcomes. However, no data have been published so far on the prevalence of TTs in ILD. A prospective, observational, cohort study was conducted within the ILD Program at the IRCCS Humanitas Research Hospital (Milan, Italy) between November 2021 and November 2023. TTs were selected according to recent literature and assigned during multidisciplinary discussion (MDD) to one of the following categories: pulmonary, etiological, comorbidities, and lifestyle. Patients were further divided into four groups according to their post-MDD diagnosis: idiopathic ILD, sarcoidosis, connective tissue disease–ILD, and other ILD. The primary study outcome was the prevalence of each TT in the study population. A total of 116 patients with ILD [63.9% male; median (IQR) age: 69 (54–78) years] were included in the study. All the TTs identified in the literature were found in our cohort, except for intractable chronic cough. We also recognized differences in TTs across the ILD groups, with less TTs in patients with sarcoidosis. This analysis provides the first ancillary characterization of TTs in ILD patients in a real setting to date.

## 1. Introduction

Interstitial lung diseases (ILD) encompass a wide spectrum of complex and heterogeneous diseases, with different factors, not limited to the lung, influencing disease progression as well as patients’ symptoms, quality of life, and ultimately survival [1,2,3,4]. Different limitations related to ILD diagnostic ontology due to the significant heterogeneity of disease behavior across ILD subtypes are currently recognized, as well as overlapping features and, thus, treatment approaches [4,5,6,7]. This heterogeneity represents a substantial barrier to understanding disease mechanisms and developing effective and personalized treatments. Precision medicine models of care are needed to improve outcomes for patients with ILD.

A “treatable trait (TT)” strategy was recently proposed as a precision medicine model of care to improve the management of ILD patients [8,9,10]. This strategy encompasses the multidimensional assessment of every patient to identify specific TTs leading to a type of management tailored to specific TTs [8,9,10]. A “TT” personalized strategy is developed around the identification of a validated biomarker that could help in understanding the biological mechanism (endotype) as well as the clinical, radiological, and functional manifestation (phenotype) of the disease [11]. Experts suggested different TTs to be identified in ILD patients across four domains [8,9]. Indeed, this strategy relies on the identification and management of TTs that are not restricted to the pulmonary domain [8]. The detection of extra-pulmonary comorbidities as well as lifestyle and behavioral TTs, although not strictly biologically related to ILDs, might impact patients’ quality of life and clinical outcomes [9,10].

The identification of TTs has led to the adoption of different and specific therapeutic strategies for several respiratory diseases that are radically different from ILD (such as chronic obstructive pulmonary disease—COPD—and asthma) [12,13]. Data on asthma patients showed that a “holistic” approach based on the identification of treatable traits might improve relevant clinical outcomes if compared to a “guideline-based” approach. [13].

Although this approach seems to be intriguing, no data have been published so far regarding the prevalence of TTs in ILD patients as well as how they change over time in a real-life scenario.

To address this knowledge gap, the objective of our analysis was to determine the prevalence and characteristics of TTs in ILD patients assessed during a well-structured multidisciplinary discussion (MDD) at a referral ILD center in Italy. The prevalence and characteristics of TTs were also assessed after 1 year to determine their longitudinal changes.

## 2. Results

### 2.1. Study Population and Study Groups

A total of 116 ILD patients [63.9% male; median (IQR) age: 69 (54–78) years] were included in the study. The median (IQR) FVC% predicted value was 81.5 (67.0–95.0), whereas the median (IQR) DLCO% predicted value was 59 (45.5–75.0). In total, 61.2% (*n* = 71) of the patients were current or former smokers. The entire characteristics of the study population are reported in Table 1. In this cohort, 41.4% (*n* = 48) of the patients were classified in the idiopathic group, 15.5% (*n* = 18) in the sarcoidosis group, 22.4% (*n* = 26) in the CTD group, and 20.7% (*n* = 24) in the miscellaneous group.

### 2.2. Baseline Prevalence of TTs

The prevalence rate of each TT in the total and domain-arrayed study population is reported in Figure 1. All the TTs identified in the literature were found in our cohort, except for intractable chronic cough. The five most prevalent TTs were CTD (*n* = 30, 25.9%), PH (*n* = 26, 22.4%), obstructive disease/emphysema (*n* = 19, 16.4%), chronic respiratory failure (*n* = 18, 15.5%), and PPF (*n* = 17, 14.7%). The less prevalent TTs were genetic (*n* = 1, 0.9%), neutrophilic inflammation (*n* = 2, 1.7%), and chronic infection (*n* = 2, 1.7%).

### 2.3. TTs in the Study Group

The median (IQR) number of TTs in each patient was two (1.0–3.8); 25% (*n* = 29) of the patients had at least four TTs (Table 2).

Patients with sarcoidosis had a significantly lower number of TTs (zero [IQR 1–2]) compared to patients in the idiopathic (two [IQR 1–4]), CTD (two [IQR 2–3]), and miscellaneous groups (two [IQR 1–5]), with *p* = 0.004, *p* = 0.001, and *p* = 0.015, respectively (Table 3). Of note, no etiological TTs were identified in the sarcoidosis group, whereas all CTD-ILD patients had at least one etiological TT.

### 2.4. TT Modification over Time

A total of 79 patients were enrolled before December 2022, and 61 were followed up for at least 12 months, as of the remaining, 9 died, 6 moved to another ILD Program, and 4 were lost to follow-up. The patients who died during follow-up had more TTs at baseline in comparison to those who survived (six [IQR 4.5–7.5] vs. two [IQR 1–3], *p* < 0.001). The median number of TTs for each patient was similar between baseline (two [IQR 1–3]) and 1-year follow-up (two [IQR 1–3]), *p* = 0.920. Analyzing the data from the 61 patients with follow-up data, the modification of some TTs was evident. Some TT increased in prevalence, such as CTD (from 16.7% [*n* = 10] to 30% [*n* = 18]), osteoporosis (from 10% [*n* = 6] to 21.7% [*n* = 13]), and side effects of treatment (from 8.3% [*n* = 5] to 13.3% [*n* = 8]). Other TTs remained stable, such as PPF (18.3% [*n* = 11] at baseline and 1-year follow-up) and PH (28.3% [*n* = 17] at baseline and 1-year follow-up). Finally, some TTs slightly decreased in prevalence, such as chronic respiratory failure (from 18.3% [*n* = 11] to 10% [*n* = 6]).

## 3. Discussion

To the best of our knowledge, this is the first report presenting data concerning the prevalence, type, and changes over time of TTs in ILD patients followed up within a multidisciplinary team (MDT) program. No data have been published so far on ILD, although some data were published on chronic obstructive pulmonary diseases [14,15]. All the TTs proposed and hypothesized in the literature in patients with ILD were found in our cohort, except for intractable chronic cough [8,9,10]. Although pulmonary TTs accounted for 5 of the 10 most prevalent TTs in our cohort, the majority of the TTs identified belonged to the comorbidity domain. In addition, we identified a high proportion of patients with the CTD (*n* = 30, 25.9%), PH (*n* = 26, 22.4%), and PPF (*n* = 17, 14.7%) traits, which underlines the importance of a multidisciplinary approach to ILD. The 2018 guidelines on IPF endorsed by the American Thoracic Society (ATS), the European Respiratory Society (ERS), the Japanese Respiratory Society (JRS), and the Latin American Thoracic Society (LATS) identified pulmonologists, radiologists, and a pathologist as the core in MDD discussion [16]. The guidelines suggest the incorporation of a rheumatologist on a case-by-case basis. Similarly, recent guidelines on hypersensitivity pneumonitis (HP) of the American College of Chest Physicians identified a pulmonologist, a chest radiologist, and a pathologist as the core of the MDT [17]. Rheumatologists and occupational medicine specialists are not considered essential components of the MDT for HP. Cardiologists are not mentioned in those guidelines. The high proportion of CTD and PH identified in our cohort suggests that cardiologists and rheumatologists should be included within the essential components of the MDT, as their role is not limited to the diagnosis but also to the management of ILD patients. Moreover, recent data suggest that some phenotypes could be identified and treated with selective drugs regardless of the ILD diagnosis, such as PPF and PH-ILD [3]. PPF refers to a spectrum of ILDs that share a phenotype characterized by an increasing extent of fibrosis on HRCT, a decline in lung function, and worsening symptoms. Nintedanib, an intracellular tyrosine kinase inhibitor with antifibrotic properties, reduces the rate of disease progression irrespective of the underlying ILD diagnosis [6]. Similarly, inhaled treprostinil was shown to improve exercise capacity, as assessed by the 6 min walk test, compared with placebo in patients with PH due to ILD [7]. Thus, the identification and management of these traits within the MDD are of paramount importance.

Although ILDs display a phenotype characterized by fibrotic lesions in the lung leading to a restrictive pattern, 16.4% of the patients in our cohort had an obstructive pattern or associated emphysema at the HRCT scan and, thus, might benefit from bronchodilators. This is the case for IPF patients with extended emphysematous lesions on the HRCT scan, named “combined pulmonary fibrosis and emphysema” (CPFE) [18,19]. These patients are characterized by more severe outcomes, including mortality [19,20]. It is well known that CPFE patients may experience acute exacerbations in the form of either IPF or obstructive disease that have important implications for both long- and short-term prognosis [21]. Moreover, some ILDs can manifest both at a bronchial and at a parenchymal level, as in the case of sarcoidosis. Granulomas related to sarcoidosis are frequently associated with airway obstruction leading to a more pronounced decline of the lung function [22,23]. Thus, bronchodilators might represent a relevant treatment in the case of coexisting emphysema or obstruction.

Some traits have a lower prevalence in ILDs compared to other respiratory diseases, such as eosinophilic inflammation, which can be found in approximately 30% of both asthma and chronic obstructive pulmonary disease patients, whereas 2.6% of ILD patients display this TT [14]. Although the long-term use of steroids and other immunosuppressive drugs is a predisposing factor for chronic infection, this trait can be detected in less than 2% of ILD patients. The chronic infection TT is more often displayed in patients with airway disease, in particular bronchiectasis patients, compared to ILD patients [15].

We also recognized differences in TTs across the ILD groups, with fewer TTs in patients with sarcoidosis compared to patients with other ILD types. Several reasons might explain this finding. First, the exact cause of sarcoidosis is not known. Although genetic susceptibility, environmental factors, putative antigens, and autoimmunity have been hypothesized in the development of this disease, no single cause has been identified to date [24]. Second, sarcoidosis occurs in younger patients compared to other types of ILDs [25]. Thus, the number of comorbidities is generally lower. Third, sarcoidosis is highly heterogeneous, with variable presentation, severity, and evolution. A high proportion of patients are asymptomatic or mildly symptomatic.

Of note, we observed an increased prevalence of baseline TTs in patients that died within the 1-year follow-up, suggesting an association between the total number of TTs and disease severity. Although the median number of TTs for each patient was similar at baseline and 1-year follow-up, relevant changes in some TTs were identified. This might be related to the beneficial effect of treatment, as suggested by the reduced prevalence of chronic respiratory failure (from 18.3% [*n* = 11] to 10% [*n* = 6]), but also to some detrimental aspects such as the increased prevalence of both osteoporosis (from 10% [*n* = 6] to 21.7% [*n* = 13]) and side effect of treatment (from 8.3% [*n* = 5] to 13.3% [*n* = 8]). The modification of these TTs, along with the increasing recognition of CTD as an etiology at 1-year follow-up, emphasizes once again the role of the MDT in ILD management. It is well known that ILDs may evolve within weeks, months, or years. The longitudinal modifications of TTs might lead to different management strategies.

While innovative, our data have strengths and limitations. To the best of our knowledge, this is the first study reporting original data concerning TTs in ILDs. Our comprehensive longitudinal analysis attempted to identify TTs in a cohort of patients belonging to a referral ILD. TT identification relied on previous literature and was validated by a well-structured MDT including several healthcare professionals. Thus, our analysis could be replicated for other ILD cohorts. Among the study limitations, the data were obtained from a small population-based prospective cohort, with potential selection bias limiting the generalizability of our findings. For example, CTD was the most common TT identified, which can represent a peculiarity of our cohort. Second, the PM_10_ and NO_2_ concentrations vary widely according to several factors, even in the same urban or rural area. Individual exposure to air pollution is difficult to assess without artificial intelligence methods; thus, we decided not to evaluate the exposure to air pollution as a TT in our cohort [26]. Third, the follow-up period was also relatively short, limiting our understanding of the impact of profile migration in the course of the disease. Fourth, we decided not to integrate patients’ preferences and values into the TT approach. Evidence from COPD demonstrated that when patients are included into the decision-making process, their outcomes significantly improve [27]. Fifth, the present study did not aim to explore the effectiveness of a TT strategy for ILD. Future randomized controlled trials comparing a TT strategy with usual care based on guideline suggestions are needed to test the effectiveness of this strategy for patients with ILD in relation to relevant outcomes, including mortality. Furthermore, it is unknown if this strategy will be cost-effective.

## 4. Materials and Methods

### 4.1. Study Design and Study Population

A prospective, observational, cohort study was conducted within the ILD Program at the IRCCS Humanitas Research Hospital, Milan, Italy. Consecutive ILD patients newly referred to the Program were recruited between November 2021 and November 2023. The inclusion criteria were age ≥18 years and a diagnosis of any ILD. Patients were excluded in the case they refused or withheld consent. The present study was approved by the local ethics committee (protocol number 857/21).

### 4.2. TT Identification

TTs were selected according to recent literature and assigned to one of the following categories: pulmonary, etiological, comorbidities, and lifestyle [8,9,10]. Pulmonary TTs include progressive fibrosis; eosinophilic inflammation; neutrophilic inflammation; acute exacerbation; acute infection; chronic infection or recurrent infection; chronic respiratory failure (CRF); intractable chronic cough; chronic breathlessness syndrome; and emphysema/obstructive ventilatory defects. Etiological TTs include connective tissue disease (CTD)s/vasculitis; drugs; exposure-related (both organic and inorganic); and genetic. Comorbidities TTs include gastro-esophageal reflux disease (GERD); pulmonary hypertension (PH); ischemic heart disease; congestive heart failure; obstructive sleep apnea (OSA); lung cancer; diabetes; osteoporosis/osteopenia; pulmonary embolism; obesity; cachexia/malnutrition; frailty; and anxiety/depression. Lifestyle TTs include smoking; adherence to treatments; side effects of treatment; lack of exercise/deconditioning of skeletal muscle; diet; and family and social support. The definition of each TT and the assessment tool to identify it were established according to recent literature [8,9,10].

The ILD patients were managed within the Program based on standard operating procedures and according to current international guidelines [28,29]. TTs were identified during pulmonary medical consultations and tabulated for each patient. A committee (F.A., A.S., and S.A.) recognized TTs considered relevant for every patient and discussed them within the multidisciplinary team (MDT) to reach a final agreement.

A standard case report form (CRF) was used to discuss each ILD case within the MDT. The CRF included a complete medical history as well as clinical examination data, laboratory test results, pulmonary function test results, high-resolution computed tomography (HRCT) images, bronchoalveolar lavage (BAL) counts, surgical lung biopsy results, and the TTs identified. The MDD was led by two pulmonologists (A.S. and F.A.) with experience in ILD. An MDD was arranged every week and lasted for about 60 min. All MDDs were attended by at least one thoracic radiologist, one thoracic pathologist, one rheumatologist, two senior pulmonologists, and a respiratory physiotherapist. A cardiologist, an oncologist, a gastroenterologist, or a nephrologist was also involved when required.

Patients were further divided into 4 groups according to the post-MDD diagnosis: idiopathic ILD, sarcoidosis, CTD-ILD, and other ILD (miscellaneous).

The TTs were reassessed during pulmonary medical consultations at 6-month and 1-year follow-ups.

The global picture of the study is presented in the graphical abstract.

### 4.3. Outcomes

The primary study outcome was the prevalence of each TT in the study population. The secondary outcomes included the prevalence of TTs among the study groups and changes in TT prevalence at 1-year follow-up.

### 4.4. Statistical Analysis

The qualitative variables were summarized with absolute and relative (percentage) frequencies. The quantitative variables were summarized with means (standard deviations, SD) and medians (interquartile ranges, IQR) according to their normal vs. non-normal distribution. The qualitative variables were compared with the chi-squared and Fisher exact tests, when appropriate. ANOVA and Kruskal–Wallis test were used to compare quantitative variables having normal and non-normal distributions, respectively. Sidak correction was adopted for multiple comparisons. Independent sample *t*-tests were used to determine the differences in the total number of TTs among the cohorts. A two-tailed *p*-value less than 0.05 was considered statistically significant. The statistical analysis was performed using SPSS, version 21.0 (SPSS, Chicago, IL, USA).

## 5. Conclusions

In conclusion, this analysis provides the first ancillary characterization of TTs in ILD patients in a real setting to date. Several TTs have been hypothesized in the literature, and all of them, except for intractable chronic cough, were identified in our cohort of ILD patients, corroborating the importance of a TT approach in ILD management. The crucial role of the MDT in the clinical management of ILD patients was confirmed by our data, showing that the majority of TTs identified belonged to the comorbidity domain and, thus, were not restricted to the lung. Moreover, we demonstrated a change in TTs at 1-year follow-up, underlying the importance of a dynamic approach. Longitudinal studies on international and multicenter cohorts are needed to confirm our findings. The cost-effectiveness of a TT strategy and the impact of this approach on clinically relevant outcomes should be tested in future randomized control trials.

## Figures and Tables

**Figure 1 biomedicines-12-01047-f001:**
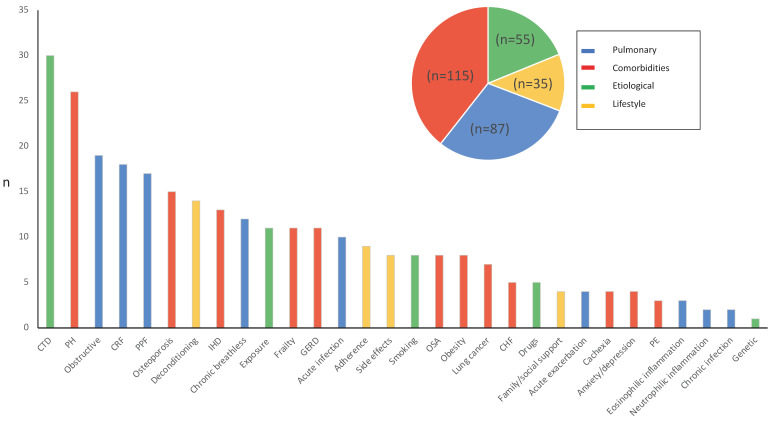
Prevalence of TTs in the study population. The pie chart reports the total number of TTs in the study population according to the TT domain (pulmonary, comorbidities, etiological, lifestyle). In the vertical Y-axis, the absolute number of each trait is reported. Each color identifies the domain of the TT identified. **Legend:** CTD: connective tissue disease; PH: pulmonary hypertension; CRF: chronic respiratory failure; PPF: progressive pulmonary fibrosis; IHD: ischemic heart disease; GERD: gastroesophageal reflux; OSA: obstructive sleep apnea; CHF: chronic heart failure; PE: pulmonary embolism.

**Table 1 biomedicines-12-01047-t001:** Characteristics of the study population.

Variables	Study Population (*n* = 116)
**Demographics**	
Female sex, *n* (%)	43 (37.1)
Caucasian, *n* (%)	112 (96.6)
Age, median (IQR)	69 (54–78)
Smoker, *n* (%)	71 (61.2)
Current	9 (7.8)
Former	62 (53.4)
**ILD diagnosis**	
Idiopathic, *n* (%)	48 (41.4)
IPF, *n* (%)	14 (12.1)
CTD-ILD, *n* (%)	26 (22.4)
Sarcoidosis, *n* (%)	18 (15.5)
Other, *n* (%)	24 (20.7)
**Onset of disease**	
<1 year, *n* (%)	39 (33.6)
1–5 years, *n* (%)	50 (43.1)
>5 years, *n* (%)	27 (23.3)
**Comorbidities**	
GERD, *n* (%)	20 (17.2)
Cardiovascular diseases, *n* (%)	72 (62.1)
Congestive heart failure, *n* (%)	9 (7.8)
Pulmonary hypertension, *n* (%)	18 (15.5)
Osteoporosis, *n* (%)	8 (6.9)
Depression/anxiety, *n* (%)	7 (6)
History of neoplastic disease, *n* (%)	29 (25)
Chronic renal failure, *n* (%)	16 (13.8)
CTD, *n* (%)	31 (26.7)
**Respiratory comorbidities**	
OSAS, *n* (%)	8 (6.9)
COPD, *n* (%)	10 (8.6)
Asthma, *n* (%)	5 (4.3)
**Functional evaluation**	
FVC, median (IQR)	81.5 (67–95)
FEV1, median (IQR)	86 (71–98)
DLCO, median (IQR)	59 (45.5–75)
DLCO/Va, median (IQR)	80 (70.5–95.5)
Any history of exposure, *n* (%)	51 (44)
Any history of use of drugs with potential lung injury, *n* (%)	31 (26.7)
**Symptoms**	
Cough, *n* (%)	58 (50)
Mucus production, *n* (%)	22 (19)
Dyspnea, *n* (%)	83 (71.6)
**Physical examination**	
Velcro-like crackles, *n* (%)	61 (52.6)
Wheezing, *n* (%)	16 (13.8)
Digital clubbing, *n* (%)	9 (7.8)
Raynaud phenomenon, *n* (%)	18 (15.5)
**Radiology**	
UIP definite, *n* (%)	15 (12.9)
UIP probable, *n* (%)	18 (15.5)
UIP indeterminate, *n* (%)	13 (11.2)
Alternative, *n* (%)	70 (60.3)
**Diagnostic work-up performed**	
Autoimmunity tested, *n* (%)	77 (66.4)
Bronchoscopy, *n* (%)	41 (35.3)
Biopsy, *n* (%)	24 (20.7)
**Chronic treatment**	
Antifibrotics, *n* (%)	13 (11.2)
Steroids, *n* (%)	42 (36.2)
Immunosuppressive, *n* (%)	33 (28.4)
Oxygen, *n* (%)	26 (22.4)
Inhalation drugs, *n* (%)	18 (15.5)
Rehabilitation, *n* (%)	8 (6.9)
Exercise training/airway clearance, *n* (%)	20 (17.2)

**Legend:** IQR: interquartile ranges; ILD: interstitial lung disease; IPF: idiopathic pulmonary fibrosis; CTD: connective tissue disease; GERD: gastroesophageal reflux disease; OSAS: obstructive sleep apnea syndrome; COPD: chronic obstructive pulmonary disease; FVC: forced vital capacity; FEV1: forced expiratory volume in the first second; DLCO: diffusion lung capacity for carbon monoxide; UIP: usual interstitial pneumonia.

**Table 2 biomedicines-12-01047-t002:** Total number of TTs for ILD patient and according to each specific TT category.

	TTs			
Median number, (IQR)	2 (1–3.75) 21 (18.1) 15 (12.9) 35 (30.2) 16 (13.8) 29 (25)			
0, *n* (%)				
1, *n* (%)				
2, *n* (%)				
3, *n* (%)				
4+, *n* (%)				
**TT category**	**Etiological**	**Pulmonary**	**Comorbidities**	**Lifestyle/** **Behavioral**
0, *n* (%)	71 (61.2)	67 (57.8)	46 (39.7)	80 (69)
1, *n* (%)	45 (38.8)	26 (22.4)	40 (34.5)	30 (25.9)
2+, *n* (%)	0 (0)	12 (19.8)	30 (25.9)	6 (5.1)

**Legend.** TTs: treatable traits.

**Table 3 biomedicines-12-01047-t003:** TTs according to the study groups.

TTs	Idiopathic (*n* = 48)	Sarcoidosis (*n* = 18)	CTD (*n* = 26)	Other (*n* = 24)	*p* Value
Median number, (IQR)	2 (1–4)	0 (0–2)	2 (2–3)	2 (1–5)	0.09 *
0, *n* (%)	6 (12.5)	10 (55.6)	0 (0)	5 (20.8)	
1, *n* (%)	9 (18.8)	1 (5.6)	2 (7.7)	3 (12.5)	
2, *n* (%)	13 (27.1)	4 (22.2)	13 (50)	5 (20.8)	
3, *n* (%)	7 (14.6)	2 (11.1)	6 (23.1)	1 (4.2)	
4+, *n* (%)	13 (27.1)	1 (5.6)	5 (19.2)	10 (41.7)	
**At least one TT for each category**					
Etiological, *n* (%)	6 (12.5)	0 (0)	26 (100)	14 (58.3)	<0.001 *
Pulmonary, *n* (%)	24 (50)	4 (22.2)	4 (23.1)	15 (62.5)	0.003 *
Comorbidities, *n* (%)	32 (66.7)	8 (44.4)	19 (73.1)	11 (45.8)	0.045 *
Lifestyle/Behavioral, *n* (%)	16 (33.3)	4 (22.2)	8 (30.8)	8 (33.3)	0.924

* **Median number of traits:** sarcoidosis vs. idiopathic, *p* = 0.004; sarcoidosis vs. CTD, *p* = 0.001; sarcoidosis vs. other, *p* = 0.015. **Legend.** TTs: treatable traits; CTD: connective tissue disease; IQR: interquartile ranges.

## Data Availability

Dr. Francesco Amati has full access to all the data in the study and takes responsibility for the integrity of the data and the accuracy of the data analysis.

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
