# Peer review of "First Report of the Prevalence at Baseline and after 1-Year Follow-Up of Treatable Traits in Interstitial Lung Diseases"

_biomedicines, 2024, doi:10.3390/biomedicines12051047_

Round 1

Reviewer 1 Report

Comments and Suggestions for Authors

The authors process data on the prevalence of TT in IBD, through a prospective, observational, cohort study conducted in the IRD program of the IRCCS Humanitas Research Hospital, within admitted patients for two years, 2021-2023.

Patients were further subdivided into 4 groups according to post-MDD diagnosis: idiopathic ILD, sarcoidosis, connective tissue disease-ILD, and other ILD. The primary outcome of the study was the prevalence of each TT in the study population. A total of 116 patients with IBD [63.9% male; median (IQR) age: 69 (54–78) years] were included in the study. Differences in TTs were determined in the ILD groups with fewer TTs in sarcoid patients. Overall, the authors make an ancillary characterization of TTs in patients with ILD in a real-world setting.

The analysis is entirely statistically based, and yet it is not clear from the discussion what exactly is innovative. The introduction part is poor, booring and should be supplemented. Schematic overview, as additional plus-part in introduction, will also be appropriate. All tables are boring, with an uncharacteristic font. The addition of a long-rank test or Cox regression or Pearson correlation will be appropriate. The analysis used in Figure 1 is not appropriate. It is incomprehensible.The discussion should emphasize what is innovative. which will be used in the future cohort of similar cases.

Comments on the Quality of English Language

Minor editing of English language required

Author Response

Reviewer 1:

General comment: The authors process data on the prevalence of TT in IBD, through a prospective, observational, cohort study conducted in the IRD program of the IRCCS Humanitas Research Hospital, within admitted patients for two years, 2021-2023. Patients were further subdivided into 4 groups according to post-MDD diagnosis: idiopathic ILD, sarcoidosis, connective tissue disease-ILD, and other ILD. The primary outcome of the study was the prevalence of each TT in the study population. A total of 116 patients with IBD [63.9% male; median (IQR) age: 69 (54–78) years] were included in the study. Differences in TTs were determined in the ILD groups with fewer TTs in sarcoid patients. Overall, the authors make an ancillary characterization of TTs in patients with ILD in a real-world setting.

Response: We would like to thank the reviewer for her/his words resuming our work.

Comment 1:

Response to comment 1: The analysis is entirely statistically based, and yet it is not clear from the discussion what exactly is innovative.

Response to comment 1: We thank the reviewer for her/his comment. Our paper is the first report of prevalence of treatable traits in ILD. Indeed, no original data has been published so far in ILD. Some data, on larger cohorts of patients, has been published only in obstructive respiratory diseases and not in ILD. To clarify and underline that our data are the first data published in ILD concerning the treatable traits strategy, we expanded the discussion section as follows. Discussion, first line “To the best of our knowledge, this is the first report gathering data concerning the prevalence, type, and changes over time of TTs in ILD patients followed up within a multidisciplinary team (MDT) program. No data has been published so far on ILD although some data has been published on chronic obstructive pulmonary diseases”. Discussion, strengths and limitation section “To the best of our knowledge, this is the first study reporting original data concerning TTs in ILD. Our comprehensive longitudinal analysis attempts to identify TTs in a cohort of patients belonging to a referral ILD. TT identification relies on previous literature and was validated by a well-structured MDT including several healthcare professionals. Thus, our analysis could be replicated in other ILD cohorts”.

Comment 2: The introduction part is poor, booring and should be supplemented. Schematic overview, as additional plus-part in introduction, will also be appropriate.

Response to Comment 2: We thank the reviewer for her/his comment. We recognize that the introduction part should be implemented. Accordingly, we modify the text as follows “A “treatable traits (TT)” strategy has been recently proposed as a precision medicine model of care to improve the management of ILD patients [8-10]. This strategy encompasses the multidimensional assessment of every patient to identify specific TTs leading to a management tailored to specific TTs [8-10]. A “TT” personalized strategy is developed around the identification of a validated biomarker which could help in understanding the biological mechanism (endotype) as well as the clinical, radiological, and functional manifestation (phenotype) of the disease [11]. Experts suggested different TTs to be identified in ILD patients across four domains [8-9]. Indeed, this strategy relies on the identification and management of TTs that are not restricted to the pulmonary domain [8]. The detection of extra-pulmonary comorbidities as well as lifestyle and behavioral TTs, although not strictly biologically related to ILDs, might impact patient’s quality of life and clinical outcomes [9-10]. The identification of TTs has led to the adoption of different and specific therapeutic strategies in several respiratory diseases that are radically different compared to ILD (such as chronic obstructive pulmonary disease -COPD- and asthma) [12,13]. Data on asthma patients have shown a “holistic” approach based on the identification of treatable traits might improve relevant clinical outcomes if compared to a “guideline-based” approach. [13].”

Comment 3: All tables are boring, with an uncharacteristic font. The addition of a long-rank test or Cox regression or Pearson correlation will be appropriate.

Response to Comment 3: We thank the reviewer for her/his comment. We modify the structures of the tables according to reviewer suggestion. Font has been selected based on authors guidelines of the journal. Statistical analysis has been performed comparing groups. Qualitative variables were compared with chi-squared and Fisher exact tests, when appropriate. ANOVA and Kruskall-Wallis were used to compare quantitative variables having normal and non-normal distributions, respectively. Sidak correction was adopted for multiple comparisons. Independent sample t-tests were used to determine differences in the total number of TTs among cohorts.  Given the small study population, we were not able to perform a regression analysis.

Comment 4: The analysis used in Figure 1 is not appropriate. It is incomprehensible.

Response to Comment 4: We thank the reviewer for her/his comment. We recognize that this comment is important. The figure 1 represents the visual interpretation of the treatable traits. However, we recognize that the figure is difficult to be interpreted. For these reasons we modify the figure that is composed of two parts: a pie chart in which are reported the total number of the TTs in the study population according to the TT domain (pulmonary, comorbidities, etiological, lifestyle). The second part of the figure includes a histogram. In the vertical Y-axis of the histogram is reported the absolute number of each trait. Each color identifies the domain of the TT identified.

Comment 5: The discussion should emphasize what is innovative. which will be used in the future cohort of similar cases.

Response to Comment 5: We thank the reviewer for her/his comment that is integrative of comment 1. The modification of the text according to both comments is reported in “response to comment 1”.

Reviewer 2 Report

Comments and Suggestions for Authors

The authors have provided results froma study on ILD claiming to report the first summary of prevalance and change over the time.

While the report is novel, I have the following concerns that addressing them can improve the paper:

1. I think the term "change over the time" is very vague. It is misleading in the title and difficult to understand in the text. I hope author come with a better term here.

2. Figure 1 was not clear as I did not see any plot, please check this figure.

3. The conclusion does not shed light on the findings of the study and can be improved.

4. The paper requires a figure to show the global picture of the study and mention the longitudinal aspects of the study.

Comments on the Quality of English Language

The text needs to be checked for minor errors and typos.

Author Response

Reviewer 2

General comment: The authors have provided results from a study on ILD claiming to report the first summary of prevalence and change over the time. While the report is novel, I have the following concerns that addressing them can improve the paper.

Response: We would like to thank the reviewer for her/his nice words.

Comment 1: I think the term "change over the time" is very vague. It is misleading in the title and difficult to understand in the text. I hope author come with a better term here.

Response to comment 1: We thank the reviewer for her/his comment. We recognize that this comment is important. We recognize that the title could be confusing. For this reason, we modify the title as follows “First report of prevalence at baseline and after 1-year follow-up of treatable traits in interstitial lung diseases”.

Comment 2: Figure 1 was not clear as I did not see any plot, please check this figure.

Response to comment 2: We thank the reviewer for her/his comment. We thank the reviewer for her/his comment. We recognize that this comment is important. The figure 1 represents the visual interpretation of the treatable traits. However, we recognize that the figure is difficult to be interpreted. For these reasons we modify the figure that is composed of two parts: a pie chart in which are reported the total number of the TTs in the study population according to the TT domain (pulmonary, comorbidities, etiological, lifestyle). The second part of the figure includes a histogram. In the vertical Y-axis of the histogram is reported the absolute number of each trait. Each color identifies the domain of the TT identified.

Comment 3: The conclusion does not shed light on the findings of the study and can be improved.

Response to comment 3: We thank the reviewer for her/his comment. We recognize that this comment is important. According to reviewer’ suggestion, we modify the conclusions as follows “In conclusion, this analysis provides the first ancillary characterization of TTs in ILD patients in a real setting to date. Several TTs have been hypothesized in literature and all of them, except for intractable chronic cough, have been identified in our cohort of ILD patients, corroborating the importance of a TT approach in ILD. The crucial role of the MDT in the clinical management of ILD patients has been confirmed by our data, showing that the majority of TTs identified belonged to the comorbidity domain and, thus, are not restricted to the lung. Moreover, we demonstrate a change in TT at the 1-year follow-up, underlying the importance of a dynamic approach. Longitudinal studies on international and multicenter cohorts are needed to confirm our findings. The cost-effectiveness of a TT strategy and the impact of this approach on clinically relevant outcomes should be tested in future randomized control trials.”

Comment 4: The paper requires a figure to show the global picture of the study and mention the longitudinal aspects of the study.  

Response to comment 4: We thank the reviewer for her/his comment. According to reviewer’ suggestion we added a graphical abstract in the supplementary materials showing the global picture of the study.

Round 2

Reviewer 1 Report

Comments and Suggestions for Authors

-

Author Response

We thank the reviewer for her/his relevant and thoughtful comments.

Reviewer 2 Report

Comments and Suggestions for Authors

The authors have improved the paper. Figure 1 still is not clear to me. I think in the generated PDF version there is a problem with figure 1. Please check it.

Comments on the Quality of English Language

It is now in a good shape.

Author Response

We thank the reviewer for he/his comment and appreciation of our work.

We checked the figure. The figure 1 reports the prevalence of TTs in the study population. It is composed of two elements, a pie chart and a histogram. In the pie chart are reported the total number of the TTs in the study population according to the TT domain (pulmonary, comorbidities, etiological, lifestyle). A different color has been assigned to each domain. The second element is a histogram. In the vertical Y-axis is reported the absolute number (not the percentage) of each trait. Each color identifies the domain of the TT identified.